# Clinical Utility of Trabecular Bone Score in Gastroenterology: A Narrative Review

**DOI:** 10.3390/biomedicines13061331

**Published:** 2025-05-29

**Authors:** Ivna Olic, Piero Marin Zivkovic, Ivana Tadin Hadjina, Ivan Zaja

**Affiliations:** Gastroenterology and Hepatology Department, Internal Medicine Department, University Hospital of Split, Spinciceva 1, 21000 Split, Croatia; piero.zivkovic@gmail.com (P.M.Z.); ihtadina@gmail.com (I.T.H.); ivan.zaja@yahoo.com (I.Z.)

**Keywords:** osteoporosis, trabecular bone score (TBS), bone mineral density (BMD), gastrointestinal diseases, inflammatory bowel disease (IBD)

## Abstract

Subjects with inflammatory bowel diseases (IBDs) have a higher opportunity for fractures due to the inflammatory potential of the disorder and because of the glucocorticoid therapy that is often inevitable. The fracture risk can be assessed by dual-energy X-ray absorptiometry and can also be combined with assessing the trabecular bone score (TBS). The evaluation of the TBS offers additional advantages in particular conditions, such as glucocorticoid-induced osteoporosis, and thus optimizes the fracture risk evaluation in the IBD subject group. A limited number of studies involving TBS in other digestive diseases is unlikely to provide sufficient evidence regarding the usefulness of TBS in gastroenterology. Our aim is to review the clinical utility of TBS in digestive diseases.

## 1. Introduction

Inflammatory bowel diseases (IBDs) are a heterogeneous group of chronic and recurrent intestinal diseases composed of two major forms: Crohn’s disease (CD) and ulcerative colitis (UC) [1], presented with a relapsing and recurrent course [2]. CD features chronic transmural impaction, with interrupted inflamed lesions grasping any part of the digestive tube, characterized by intestinal granulomas, obstruction, strictures and fistulas [3], whereas UC causes continuous mucosal inflammation extending from the rectum toward the colon, but without the above involvement [4]. This debilitating condition can also occur with disorders beyond the digestive tract, known as extra-intestinal manifestations (EIMs), and the main ones are those affecting joints, causing both peripheral and axial arthropathies, skin involvement, causing erythema nodosum, aphthous ulcers, or pyoderma gangrenosum, lesions of the biliary tract, leading to primary sclerosing cholangitis, and eye involvement, causing mainly uveitis, scleritis [5].

Osteoporosis is considered to be a chronic, metabolic skeletal disease, distinguished by reduced bone mass and microarchitectural impairment of bone tissue, with a consequential increase in bone fragility and susceptibility to fracture [6]. Osteoporosis is a highly prevalent disorder considered to affect over 200 million people worldwide, causing a major international social and economic burden [7], and it is followed by a reduction in bone mineral density (BMD) [8]. Recent evidence demonstrated that pro-inflammatory cytokines act on the bone-remodeling flow and represent a pivotal role in the pathogenesis of osteoporosis [9,10]. Also, the participation of both innate and adaptive immunocytes in the development of osteoporosis has been proposed [11,12]; accordingly, osteoporosis can be understood as an inflammatory disorder. It is widely known that any chronic inflammation caused by refractory infections and autoimmune disorders, for example, rheumatoid arthritis, IBD, autoimmune thyroid disease, both type 1 diabetes and type 2 diabetes, and some viral infections (i.e., human immunodeficiency virus), is frequently associated with decreased BMD, and the higher risk of osteoporosis with co-existing progressive inflammation has been firmly established [13,14,15,16,17,18]. Also, elderly individuals with autoimmune rheumatic and gastrointestinal disorders face higher fracture risks when using proton pump inhibitors, especially when there is a simultaneous usage of glucocorticoids [19].

Recent reports have suggested a connection regarding osteoporosis and lifestyle, and dietary and genetic agents unique to particular ethnic groups [20,21,22].

Malnutrition, delayed puberty, reduced physical activity, and 25-hydroxyvitamin D (25-OHD) deficiency have also been proposed in other reports [23,24].

Different gastrointestinal conditions, such as IBD, can negatively influence bone status by deteriorating nutrient and calcium absorption and producing inflammatory factors that interfere with bone resorption and formation, consequently leading to deterioration [25].

The formation of osteoporosis and the incidence of pathological fracture in individuals with IBD has grown over the decade [26]; so, establishing a causal link between IBD and osteoporosis from an epidemiological point of view has been quite challenging.

The etiopathogenesis of skeletal manifestations in IBD is multifactorial and has not been completely explained to date. In addition to systemic inflammation, numerous agents have been determined to influence bone deterioration in individuals with IBD, such as long-term corticosteroid therapy, malabsorption and menopause [27].

Also, it is well known that the levels of many proinflammatory osteoclast activators (such as TNFα and interleukin-1β in the mucosa of the gastrointestinal tract and interleukin-6 in peripheral blood) are increased in individuals with IBD compared to healthy subjects [28].

The literature indicates that osteoporosis is present in 12–42% and osteopenia in up to 77% of individuals with IBD [29,30,31,32]. Numerous reports in adults have clearly indicated a higher risk of vertebral fractures [33,34,35].

The diagnosis of osteoporosis has been based on the BMD assessed by dual-energy X-ray absorptiometry (DXA) since the World Health Organization (WHO) defined osteoporosis diagnosis as a BMD 2.5 standard deviation or more below the average value of healthy young women (T-score) [36]. BMD is influenced by alterations in bone size and can be falsely higher by degenerative changes, showing to be a suboptimal fracture predictor [37].

DXA bone densitometry can be used not only for BMD assessment but also to precisely define bone quality by the means of trabecular bone score (TBS) and hip structural analysis measurement. TBS may elucidate and enrich skeletal information that is not measured by the standard BMD evaluation. TBS is a software-based tool for analysis of DXA images of the lumbar spine by assessing pixel gray-level variations in the image. Increased TBS levels are obtained in more homogeneously textured bone, and decreased TBS levels in less well-textured bone. TBS cut-off points for classifying normal and abnormal TBS values have not been completely proposed yet. In postmenopausal women, the following normal range was suggested: TBS > 1.350 as normal; TBS between 1.200 and 1.350 is shown to be consistent with partially degraded microarchitecture; and TBS ≤ 1.200 defines degraded microarchitecture [38].

Numerous reports have indicated that TBS is significantly linked with direct measurements of bone microarchitecture, and may be a useful adjunct to BMD for the detection and assessment of fragility fractures in primary osteoporosis [38,39,40,41]. Since BMD evaluates bone quantity and TBS evaluates bone quality, these methods have been shown to be complementary in predicting the incidence of fracture and response to treatment [42].

TBS correlates more precisely with trabecular microarchitecture than BMD when used in premenopausal women and in men with idiopathic osteoporosis and low-traumatic fractures [43].

Interestingly, in particular conditions, such as glucocorticoid-induced osteoporosis and in diabetes mellitus, TBS appears to out-perform DXA [44].

TBS has been shown to be an BMD-independent clinical risk tool for developing fragility fracture and could determine individuals who have an enlarged risk of fragility fractures better than BMD [45,46].

Numerous recent studies, as well as meta-analyses, have established the value of TBS in BMD measurements, proposing this new parameter to be included in international guidelines [47,48].

According to the conducted studies, it has been concluded that, in dissimilar individuals with the same BMD, structural bone health is more precisely distinguished by assessing TBS [36,49,50]. The supplemented use of TBS and BMD remarkably enhances the prediction of fractures [51].

Different agents have been proposed for osteoporosis treatment, such as bisphosphonates, denosumab, selective estrogen receptor modulators, and calcitonin [52]. However, the various etiologies of osteoporosis, for example, age-related bone loss, long-term corticosteroid treatment, or chronic inflammatory diseases, challenge the efficacy of the current therapeutic agents. Nowadays, pharmacological modalities for osteoporosis, such as bisphosphonates, antibodies, parathyroid hormone (PTH)-related peptides, and teriparatide, exhibit limitations in their therapeutic efficacy and are linked with consequential side effects. So, there is an urgent need to enable comprehensive therapeutic methods connected with a favorable safety profile. Some reports show that colon-targeted engineered postbiotics, protecting the gut, can decrease chronic inflammation, which rebalances inflammation-disturbed osteoblast–osteoclast levels, resetting bone homeostasis [53].

The goal of our study is to elucidate the use of TBS in clinical practice with precise attention to fracture risk assessment, differential diagnosis and evaluation of treatment outcomes in patients suffering from gastrointestinal diseases.

## 2. Materials and Methods

A comprehensive literature search was conducted for articles published in MedLine via PubMed using an approach similar to the PRISMA guidelines. Fracture syntax comprised ‘trabecular bone score’ OR ‘TBS’ [search term(Title/Abstract)], AND ‘bone mineral density’ [search term(Title/Abstract)] OR ‘BMD’ [search term (Title/Abstract)] (Figure 1), with no time restrictions. The articles were adequate for review if they met the following general screening criteria: (i) an original, full-text report with TBS as a primary outcome, and (ii) available in the English language. Further eligibility criteria were specific to each of the four topics. A total of 136 papers were reviewed by 2 independent reviewers.

## 3. Results

A total of 136 studies met the eligibility criteria (prospective study design, conducted in men and/or women aged 18 years or over). Several of these reports were conducted in individuals with rheumatology diseases, and most of them in patients with endocrinology diseases and chronic kidney disease. A small number of studies were conducted in individuals with hematological and neurological disorders, and in those with chronic infectious diseases (AIDS, HCV).

We found only seven studies conducted in individuals with gastrointestinal disorders (Figure 1).

Our choice was to conduct a narrative review of the literature; because this study is not a systematic review and few articles were found, evaluations of methodological quality were not used to exclude papers from this study.

### 3.1. TBS and IBD

It is well known there is the discordance between low BMD and high frequency of vertebral fractures in a cohort of CD individuals [54], but there are only four studies [55,56,57,58] comparing bone quality measured by TBS in IBD subjects (Table 1).

The study of Soare included both CD and UC individuals. It is a cross-sectional, multicentric study, with 80 IBD individuals, with a positive correlation between BMI and TBS. But there is the lack of a control group and no assessment of disease duration nor of activity [55].

Krajcovicova et al. conducted a study of 50 adults only with CD, and concluded that TBS, but not BMD, was significantly influenced by the severity of CD [56].

In other study, Soare [57] provided an evaluation of disease duration and activity. This study found that decreased BMD status was more predominant in IBD individuals compared with healthy subjects. They found that inflammation activity is negatively associated with TBS in CD individuals. The Harvey–Bradshaw index, a marker of disease activity, is independently correlated with TBS values in CD individuals, while exposure to high-dosage glucocorticoids has an adverse effect on both BMD and TBS values [57]. The limitation of this study is that IBD individuals were not treated with vitamin D supplements, with their mean vitamin D levels being borderline between insufficiency and deficiency, which could have impact on the TBS values. Further prospective studies should be conducted with a prospective follow-up of TBS changes in comparison with BMD changes during different stages of the inflammation and after vitamin D supplementation. Also, they did not find differences in the DXA parameters between CD and UC individuals, and the literature regarding the subject is contradictory. Ezzat [59] and Bjarnason [60] obtained reduced BMD values in CD individuals compared to UC individuals.

In a report of TBS in pediatric subjects with IBD, the TBS values in the pediatric group of patients with CD were reduced compared to healthy children of similar characteristics (age and gender). TBS was significantly decreased in individuals with CD compared to in those with UC. TBS showed a correlation with BMI Z-score, the phosphorus level at diagnosis, and with age at the time of DXA scan [61].

Haschka et al. [58], in their study, evaluated TBS and BMD only in CD individuals, and the subjects were divided into subgroups according to glucocorticoid usage and disease duration. They found decreased TBS in individuals on glucocorticoid treatment.

### 3.2. TBS and Cirrhosis

Individuals with liver cirrhosis, regardless of its etiology, have an increased prevalence of osteoporosis compared to the general population [62]. There are only two reports on the link between TBS and vertebral fractures in individuals with cirrhosis, which is unusual given the common knowledge that there is a considerable fracture burden in liver transplant recipients [63].

Ogiso et al. [64], in their study, indicated that TBS can help identify individuals with cirrhosis at risk of vertebral fractures, and provided complementary information to BMD in detecting fracture incidence. In fact, their study presents the idea that TBS has a certain advantage over BMD in assessing vertebral fracture risk in cirrhotic individuals, especially when ascites is present.

In their study, Stulic et al. used a comprehensive individualized clinical fracture risk evaluation approach assessing both BMD and TBS and serum bone turnover biomarkers to compare adult male individuals with alcoholic liver cirrhosis, individuals with chronic alcohol abuse, and a healthy age- and sex-matched cohort. In their study, the TBS analysis showed that the vertebral micro-architecture was significantly preserved in the healthy cohort compared to those with alcoholic liver cirrhosis and chronic alcohol abuse [65].

Since there is a lack of studies evaluating the usefulness of TBS in cirrhotic individuals, further prospective studies are required.

An important finding of the study of Yokoyama et al. [66] was that an increased BMI or a large waist circumference was connected to the preservation of BMD in individuals with metabolic-dysfunction-associated steatotic liver disease (MASLD), which is consistent with the phenomenon reported in the literature as the “obesity paradox” [67]. Maybe future trials evaluating TBS instead of only BMD would indicate that there is bone deterioration in these individuals.

### 3.3. TBS and Chronic Atrophic Gastritis

Chronic atrophic gastritis (CAG) is a long-term inflammatory condition in the gastric mucosa characterized by the deterioration of regular glandular structures and replacement with connective tissue or non-native epithelium [68], and CAG leading to the malabsorption of nutrients, meaning a reduction in vitamin D and calcium absorption [69], consequently causing bone deterioration [70]. Common etiologies of CAG include autoimmunity and *Helicobacter pylori* infection [71]. *H. pylori*-related chronic gastritis enhances the risk of osteoporosis, with higher activity of gastritis and more extensive atrophy leading to further enhanced risk [72]. *H. pylori* infection has been shown to be inversely correlated with vitamin D status, and decreased vitamin D levels lead to a higher risk of fracture [73]. There is only one study exploring using TBS to detect impaired skeletal health in individuals with CAG, and with quite a small number of participants. In spite of those limitations, Aasarød et al. [74] assessed BMD and bone quality in individuals with CAG, and found that both BMD and TBS were decreased in the CAG cohort compared to the healthy subjects.

## 4. Discussion

Our comparative analysis of the literature about the usage of TBS in gastrointestinal diseases underlined the need for further studies.

BMD is a standard and widely accepted tool for diagnosing osteoporosis and predicting fracture risk. However, research suggests that BMD alone cannot accurately assess bone strength [75], and in order to improve the assessment of bone strength, TBS has been utilized for years [76].

Knowledge regarding the connection between decreased TBS levels in individuals with endocrine and rheumatological diseases is well elucidated in the literature. On the other hand, when it comes to conditions affecting the gastrointestinal tract, the discourse is undoubtedly more complex and requires further studies.

In the gastrointestinal spectrum of diseases, TBS was found to be efficient in the field of IBD individuals. But there is a high heterogeneity between the reports, and differences in study characteristics; so, there is a need for future trials.

Lower BMD is one of the most common extraintestinal symptoms of celiac disease [77]. Data on the incidence of osteopenia and osteoporosis often vary between reports. In other studies, there was not any significant difference between osteopenia and osteoporosis in the femur and spine in patients with celiac disease [78]. Unfortunately, no studies reporting TBS levels in celiac disease were found.

The fracture risk in eosinophilic esophagitis (EoE) patients was not statistically significantly increased compared to that in non-EoE reference subjects according to Garber et al. [79]; however, we cannot exclude the possibility that steroid therapy among individuals with EoE is linked with a moderately increased fracture risk; so, it would be interesting to assess the TBS levels in those individuals.

The International Society for Clinical Densitometry (ISCD) proposes that TBS can be used in assessing together with BMD, though not as a stand-alone tool, to refine fracture risk evaluation [42]. Since BMD evaluates bone quantity and TBS evaluates bone quality, these measurements can be considered complementary in predicting fracture risk and response to therapy in appropriate individuals.

The literature analysis points to the higher use of TBS in clinical practice in the fields of rheumatology, endocrinology and nephrology, but there is a lack of studies using TBS as a predictive tool in gastroenterology.

## 5. Conclusions

In conclusion, it is valuable to use the DXA along with TBS analysis for improved prediction of fracture risk. TBS has potential as a clinical tool in gastroenterology; however, there is a need for future prospective studies that involve the use of TBS in individuals with gastrointestinal diseases.

## Figures and Tables

**Figure 1 biomedicines-13-01331-f001:**
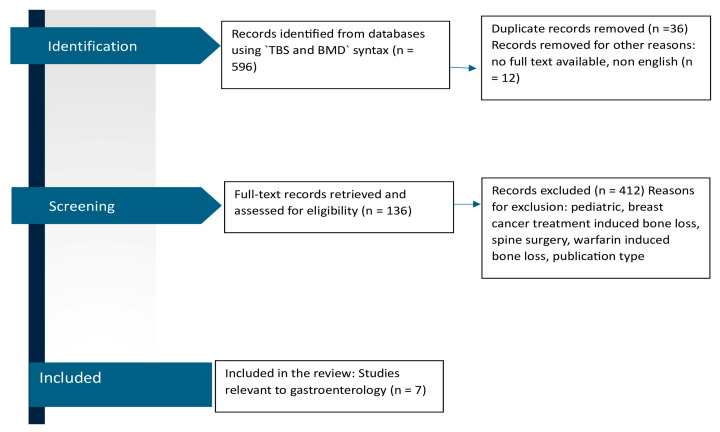
PRISMA flow diagram of the literature search process for studies investigating TBS and BMD in gastroenterology.

**Table 1 biomedicines-13-01331-t001:** Study characteristics; studies comparing bone quality measured by TBS in IBD patients.

	Design	CD *	UC ^†^	Disease Activity	Disease Duration	Subgroups According to the Treatment
Soare I, et al. [55]	Multicentric, cross-sectional	yes	yes	no	no	no
Krajcovicova A, et al. [56]	Single-center, cohort	yes	no	no	yes	yes
Soare I, et al. [57]	Single-center, cross-sectional	yes	yes	yes	yes	yes
Haschka J, et al. [58]	Single-center, cross-sectional	yes	no	no	yes	yes

* Studies evaluated patients with Crohn’s disease (CD); ^†^ Studies evaluated patients with ulcerative colitis (UC).

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
