# Peer review of "Clinical Utility of Trabecular Bone Score in Gastroenterology: A Narrative Review"

_biomedicines, 2025, doi:10.3390/biomedicines13061331_

Round 1
Reviewer 1 Report
Comments and Suggestions for Authors
The manuscript entitled” Clinical utility of Trabecular Bone Score in Gastroenterology: A narrative review” has been written well and provides a broader use of the Trabecular Bone Score in gastroenterology. However, a few things need to be updated before being accepted into this journal.
The introduction was fine.
I have a few suggestions that may help strengthen the Materials and Methods section, particularly around transparency and reproducibility:
- Clarification on Literature Search Approach: The author mentions that a “similar approach to PRISMA guidelines” was followed. To enhance the methodological rigor, I suggest clearly stating that PRISMA 2020 guidelines were used (if applicable) and including the corresponding flow diagram and citation (Page et al., 2021). This small addition can significantly boost the credibility of the review process.
- Literature Search Timeframe: While you mention “no time restrictions,” readers would benefit from knowing the specific time frame covered (e.g., from 2010 to January 2024). Even if no filters were applied, this information helps situate the scope of the search.
- Database Limitation: Reviewer noticed that only PubMed was used for the literature search. If this was intentional due to topic specificity, a brief justification would be helpful. If feasible, expanding to include other databases like Embase or Scopus could enhance comprehensiveness. The methodology currently lacks any mention of data extraction protocols or risk of bias/study quality assessment tools (e.g., GRADE, NOS). This is a critical component of systematic or structured reviews and should be addressed.
Figure 1 could benefit from a clearer and more visually engaging standard PRISMA flow diagram for consistency and readability. Additionally, it currently shows that only 7 papers were included, which seems relatively limited to draw broad conclusions or support recommendations with confidence.
I would encourage expanding the literature pool if possible and explicitly stating the inclusion/exclusion criteria used during screening. This would not only improve transparency but also strengthen the scientific basis of your conclusions.
The topic is important and shows promise in advancing clinical insight. I hope the comments provided will help strengthen your work. I look forward to seeing a revised version that addresses the suggestions and further enhances the clarity, rigor, and impact of your study.
Author Response
I agree with suggestions, and I am sending you revisted version, as you proposed.
Reviewer 2 Report
Comments and Suggestions for Authors
Congratulations to the authors on a knowledgeable review article.
The topic they address is highly relevant, especially considering the increasing number of patients with autoimmune and immune-mediated disease. The study has shown that TBS is increasingly used in clinical practice across various fields such as rheumatology, endocrinology, and nephrology, while in gastroenterology, studies evaluating its predictive value are still lacking. The study highlights the importance of assessing TBS as part of fracture risk prediction in patients with chronic diseases and emphasizes the need for further research on this topic in the context of chronic gastrointestinal diseases.
Author Response
We corrected manuscript as you suggested
